# A Review of the Measurement of the Neurology of Gait in Cognitive Dysfunction or Dementia, Focusing on the Application of fNIRS during Dual-Task Gait Assessment

**DOI:** 10.3390/brainsci12080968

**Published:** 2022-07-23

**Authors:** Sophia X. Sui, Ashlee M. Hendy, Wei-Peng Teo, Joshua T. Moran, Nathan D. Nuzum, Julie A. Pasco

**Affiliations:** 1Instiute for Mental and Physical Health and Clinical Translation (IMPACT), Deakin University, Geelong, VIC 3216, Australia; jtmoran@deakin.edu.au (J.T.M.); julie.pasco@deakin.edu.au (J.A.P.); 2Institute for Physical Activity and Nutrition (IPAN), School of Exercise and Nutrition Sciences, Deakin University, Geelong, VIC 3216, Australia; a.hendy@deakin.edu.au (A.M.H.); n.nuzum@deakin.edu.au (N.D.N.); 3Physical Education and Sports Science Academic Group, National Institute of Education, Nanyang Technological University, Singapore 308232, Singapore; weipeng.teo@nie.edu.sg; 4Department of Medicine—Western Campus, The University of Melbourne, St Albans, VIC 3010, Australia; 5Department of Epidemiology and Preventive Medicine, Monash University, Melbourne, VIC 3004, Australia; 6Barwon Health, University Hospital Geelong, Geelong, VIC 3220, Australia

**Keywords:** gait, dual-task gait, cognitive dysfunction, Alzheimer’s disease, fNIRS

## Abstract

Poor motor function or physical performance is a predictor of cognitive decline. Additionally, slow gait speed is associated with poor cognitive performance, with gait disturbances being a risk factor for dementia. Parallel declines in muscular and cognitive performance (resulting in cognitive frailty) might be driven primarily by muscle deterioration, but bidirectional pathways involving muscle–brain crosstalk through the central and peripheral nervous systems are likely to exist. Following screening, early-stage parallel declines may be manageable and modifiable through simple interventions. Gait–brain relationships in dementia and the underlying mechanisms are not fully understood; therefore, the current authors critically reviewed the literature on the gait–brain relationship and the underlying mechanisms and the feasibility/accuracy of assessment tools in order to identify research gaps. The authors suggest that dual-task gait is involved in concurrent cognitive and motor activities, reflecting how the brain allocates resources when gait is challenged by an additional task and that poor performance on dual-task gait is a predictor of dementia onset. Thus, tools or protocols that allow the identification of subtle disease- or disorder-related changes in gait are highly desirable to improve diagnosis. Functional near-infrared spectroscopy (fNIRS) is a non-invasive, cost-effective, safe, simple, portable, and non-motion-sensitive neuroimaging technique, widely used in studies of clinical populations such as people suffering from Alzheimer’s disease, depression, and other chronic neurological disorders. If fNIRS can help researchers to better understand gait disturbance, then fNIRS could form the basis of a cost-effective means of identifying people at risk of cognitive dysfunction and dementia. The major research gap identified in this review relates to the role of the central/peripheral nervous system when performing dual tasks.

## 1. Gait and Brain

Gait, defined as the manner or style of walking (e.g., typical walking speed, fast-walking, or obstacle avoidance), is an important musculoskeletal and neurophysiological phenomenon that is related to the coordination of neuronal activity in the brainstem, cerebellum and cerebrum, and spinal cord [1]. While basic gait patterns are associated with automatic rhythm activities controlled within the spinal cord, complex and dynamic gait patterns are supported and controlled by higher brain centres, including the primary motor cortex, premotor cortex, supplementary motor area, and other grey matter structures [1]. The prefrontal cortex (PFC), which is involved in executive function and is active when walking, is increasingly implicated in higher-level control of gait. Damage to the PFC can result in impaired gait, especially when performing additional complex tasks involving attention or memory during walking [1]. Epidemiological and neuropsychological evidence suggest that gait and the brain are closely related, but our current understanding of the underlying mechanisms are inadequate.

Gait disorders reflect dysfunction in the nervous system and are associated with neuromuscular diseases of the central and peripheral nervous systems [2]. Gait is recommended as a marker of brain function in many geriatric studies [3] because—unlike the brain—gait is easily observable [3]. Emerging evidence has shown that gait analysis may enable the diagnosis of diseases or disorders related to brain dysfunction, including dementia [4]. Figure 1 provides a “brain map” to visualise our proposed model in this review.

### Criteria and Searching Protocol

The initial literature search was performed in December 2021, and multiple searches were conducted subsequently up until the date of submission. This is not a systematic literature review; however, we applied the following search strategy: We used key words including “fNIRS” or “functional near infrared spectroscopy”, “gait”, “cognitive function” or “cognition” or “dementia” or “Alzheimer’s”, “dual-task”, and “neurology”. We searched the above key words in PubMed, Google Scholar, and Web of Science. This search method resulted in all available fNIRS articles related to gait and cognition. We applied the following inclusion and exclusion criteria: we included full-text articles only. Only the original studies were included in the descriptions and discussions. Relevant review articles and books were read and cited but not included in the main discussion sections. Date restrictions were set on the literature search specifying a publication date within 5 years.

## 2. Gait, Cognitive Dysfunction, and Dementia

Several measures of gait and physical capacity are used clinically, such as gait assessed at normal and fast walking speeds. Additional tests that assess walking speed and balance include the timed up and go (TUG) test [5]. Additionally, the short physical performance battery (SPPB) is a validated tool that evaluates lower-extremity physical performance using three timed tests: standing balance, walking speed, and chair stand tests [6]. Another example is a six-minute walking test [7]. The tasks are cost-effective and easy to perform in clinical settings with minimal equipment, providing high reliability and validity [8,9]. However, new and more sensitive tools and protocols are needed to detect subtle disease- or disorder-related changes in gait. Gait parameters such as stride length are important in certain neurological conditions such as Parkinson’s [10]. A small stride length may be indicative of balance issues which are implicated with cognitive deficits [11]. Additionally, while gait and balance can be considered separate constructs, they are intrinsically linked [12].

Dementia encompasses a wide range of progressive and acquired neurocognitive disorders [13]. Overall, despite being the second largest contributor to death in Australia, there are currently no effective psychological or pharmacological therapies for dementia [14]. Cognitive dysfunction, or mild cognitive impairment, is defined as diminished or impaired mental and/or intellectual functioning and is considered to be a transitional stage from normal cognitive function to dementia. It can also be regarded as a pre-dementia stage [15,16]. Cognitive dysfunction has been associated with mortality and substantially reduces quality of life [15,17,18]. Cognitive function is associated with skeletal muscle health [19]—negatively with dynapenia [20] and sarcopenia [19] and positively with handgrip strength [21], usual gait speed [21], muscle quality [22], and muscle density [23].

Slow gait speed has been associated with poor psychomotor function and attention, suggesting that it is an indicator of cognitive dysfunction [21]. Since motor decline can be more visually detectable than mild cognitive impairment, it has recently been proposed that motor function assessment be added to a comprehensive cognitive test battery for the diagnosis of Alzheimer’s disease (AD) [24].

Darweesh et al. [25] investigated the associations between gait, cognitive decline, and incident dementia in a longitudinal study. Thirty gait parameters in seven gait domains were assessed using an electronic walkway device. Poor performance in some of the independent gait domains (base of support, pace, and rhythm) were found to be predictors of cognitive decline and incident dementia. Another longitudinal study [26] followed up 144 non-dementia participants (mean age 74 years, 46% men). Gait speed was determined and recorded using an electronic walkway, and global cognitive function was assessed with the Montreal Cognitive Assessment (MoCA). Gait speed, cognition, medical status, functionality, incidence of adverse events, and dementia were recorded every two years, and 17% of participants were diagnosed with dementia during follow-up over the seven-year study. This study reported that impairments in gait domains (e.g., pace and variability) and in cognitive function were associated with an increased hazard of developing dementia, suggesting that dual declines in gait and cognition were associated with higher risk of dementia.

Previous research has used gait patterns and quantitative gait analysis [27] to identify dementia subtypes. Neurological symptoms of dementia are associated with gait characteristics such as slow speed and high variability [27], and unique spatiotemporal gait characteristics are associated with specific types of dementia [28]. Evidence shows that temporal and spatial gait parameters are more disturbed in the late stage of dementia and in non-Alzheimer’s dementia versus Alzheimer’s [4]. A multicentre cross-sectional study by Allali et al. investigated the associations between the subtype and severity of cognitive impairment/dementia, falls, and gait during progressive dementia. This study included 2496 adults across seven countries (mean age 77 years, 45% men): 1161 were cognitively healthy, 529 had mild cognitive impairment, 456 mild dementia, and 350 moderate dementia. The authors found that the association between poor gait performance and falls was moderated by cognition status (cognitive impairment subtypes), suggesting that the mechanism of falls in older people with dementia differs from that in healthy controls [29].

These studies suggest that comprehensive gait assessment, rather than the standard measurement of walking speed, is needed to identify individuals at higher risk of developing dementia. If gait is measured using a computer-based tool, and gait parameters in gait domains are categorised systematically, subtle changes in gait can be detected in a reliable and valid way.

## 3. Dual-Task Gait Performance, Cognitive Dysfunction, and Dementia

There is a need to provide a framework or model that supports the use of dual tasking as a suitable measure of cognitive decline [30,31]. Walking while performing cognitive or motor tasks (dual-task gait), such as walking while carrying an object (e.g., a shopping bag) or while talking on the phone, are part of daily life [32]. Additional dual-task conditions include backwards counting and verbal fluency-tasks, with the purpose of the secondary task being to elicit increased difficulty in the original motor task (i.e., walking or postural control) [33]. Theoretical and empirical evidence show that dual-task practice improves physical and mental performance more than single-task practice [34]. The underlying cognitive mechanism is that dual-task performance requires the allocation and rescheduling of cognitive resources for inhibition and activation of the body and brain, a process that is not required for a single task, therefore making dual tasks more difficult [34]. Dual-task gait performance is assessed when people perform two tasks (e.g., one motor task and one cognitive task) concurrently in a laboratory setting or a real-world environment [34]. Dual-task gait assessment is often included in studies of neurological disorders [35].

Single-task gait performance is considered a clinical marker of progression from mild cognitive impairment to dementia [36]. However, dual-task gait performance reflects the interaction of cognition and motor function, so it may be a more efficient predictor of this transition [36]. A cross-sectional study reported that dual-task gait performance could be used to distinguish patients with AD from those with preserved cognition or mild cognitive impairment [37]. This study included 118 older adults (40 with preserved cognition, 40 with mild cognitive impairment, and 38 with mild AD). Gait was measured by a 10-meter walk task; dual-task performance was measured by a TUG task combined with a motor cognition task (calling a phone number). No differences between the three groups were found in single-task gait performance, but the AD group had significantly poorer dual-task gait performance, while there was no difference in the other two groups. One of the limitations for this study was that gait parameters were not assessed using a gait mat or equivalents.

Montero-Odasso et al. [36] investigated whether dual-task gait performance was associated with the development of dementia in 112 older adults with mild cognitive impairment. Participants were followed-up every two years over a decade. Single- and dual-task gait performance was measured using an electronic walkway; dementia outcomes were determined using Diagnostic and Statistical Manual of Mental Disorders criteria. Poor dual-task gait performance was associated with incident all-cause dementia.

In a clinical setting, Lowe et al. [38] examined the differences between participants with mild cognitive impairment and a group with subjective cognitive complaints in a retrospective study of single-task versus dual-task gait performance (252 participants; 145 women; mean age 66 years). They reported that under dual-task conditions, participants with mild cognitive impairment walked more slowly, and their cognitive decline was greater than those with subjective cognitive impairment. In addition to demographic and health risk factors, neuropsychological measures of executive attention explained the differences in dual-task gait performance, providing evidence for the use of dual-tasks to identify mild cognitive impairment in clinical settings.

Boettcher et al. [39] developed and employed a machine learning program to assess dual-task gait performance with the aim of detecting early cognitive decline and enabling the prevention of the development of dementia. Their case–control study involved 92 participants (31 healthy controls, 61 with mild cognitive impairment). Gait performance, including dual-task gait, was recorded using an electronic walkway. The results showed that their protocol has potential for early diagnosis of cognitive decline; however, it has yet to be applied to a large study sample, and its real-world cost effectiveness remains to be established.

Varma et al. [40] examined novel gait parameters that might distinguish participants with AD from healthy members of the community. This study involved 38 participants with mild AD and 48 healthy controls. Gait performance was assessed as variability in amplitude, pace, rhythm, symmetry, and variability and was measured continuously over seven days using a portable gait recorder. The study reported that variability in step velocity and cadence is an effective predictor of AD and that those parameters are associated with cognitive impairment in AD. The authors suggested that in population-based studies, continuous gait monitoring is a valid and reliable method for identifying people at risk of developing dementia.

## 4. Neurophysiological Biomarkers

Understanding gait deficits due to functional and structural brain changes in cognitive dysfunction and dementia may provide insights into ways of delaying or preventing dementia. Dual-task gait is a better predictor of cognitive dysfunction or dementia than single-task gait [41], but neurophysiological markers of dual-task gait remain poorly understood and not often assessed. Gait deficits are associated with reduced brain volume (entorhinal cortex [42], primary motor cortex [43], grey matter [44], and hippocampus [45]). However, partially due to the high cost of specialist equipment, such as functional magnetic resonance imaging (fMRI), most research on gait deficits to date have involved small numbers of participants, giving low statistical power for detecting differences and changes over time [46].

The limitation of fMRI is that it can only be operated during a mimic gait task rather than a real gait task. Burki et al [47] designed an fMRI paradigm to examine age-related neural mechanism during gait tasks in 15 younger adults (mean age 28 years) and 31 healthy older adults (mean age 76 years). Gait performance in single and dual tasks was measured using an electronic walkway. Brain functioning and structures were scanned using fMRI, while participants performed imitated gait tasks. Activation in the primary motor cortex and in supplementary motor areas decreased more than activation in the superior parietal lobe with the switch from the single- to dual-task condition. Increased activation in the superior parietal lobe was associated with lower stepping speed and lower executive control in dual tasks, indicating the important role of the superior parietal lobe. This study demonstrates the feasibility of using fMRI to investigate brain activity during imitated walking tasks, but its expense limited the sample size and rendered it less cost effective. Portable brain imaging equipment, such as fNIRS, is needed to study brain activity during real gait tasks.

### 4.1. fNIRS, Dual-Task Gait, and Ageing

Functional NIRS is a non-invasive, cost-effective, safe, portable, and non-motion-sensitive neuroimaging technique [48]. Due to these advantages, fNIRS has been widely used in studies of clinical populations such as people suffering from AD, depression, and other chronic neurological disorders [49]. Functional NIRS maps neuronal activity by applying near-infrared-spectrum light that can travel through tissue. It is sensitive to brain neurovascular changes during neuroactivity [50]. Specifically, the near-infrared light of specific wavelengths is absorbed by oxygenated and deoxygenated haemoglobin (Hb) in the cortex. An increase in oxygenated Hb is considered an indicator of neuronal activity, and a common hemodynamic response involves an increase in oxygenated Hb accompanied by a decrease in deoxygenated Hb. Studies have shown that the outcomes from fNIRS and fMRI are highly reliable and valid, with both measuring the blood-oxygen-level-dependent signal, making fNIRS a good surrogate measure.

Beurskens et al. [51] studied age-related prefrontal activity during dual-gait tasks in 15 younger and 10 older people (mean age 25 years versus 71 years). Participants performed dual tasks including treadmill walking while simultaneously performing either a visual or verbal memory task. Motor performance and prefrontal context activity were assessed using fNIRS, revealing that prefrontal activation decreased significantly during performance of a visual task while walking for the elderly but not for the younger group. The authors argued that older people may shift from processing neural resources from the PFC to other brain regions during these types of higher-order dual cognitive–motor tasks. This might align with the HAROLD hypothesis [52]—Hemispheric asymmetry reduction in older adults—whereby, with ageing, more diffuse brain regions are utilised for a task when compared to younger adults that have more localised brain activation.

Chen et al. [53] investigated the role of the PFC in obstacle negotiation during dual-task gait. This study included 90 older adults without dementia (mean age 78 years, 39 men). Functional NIRS was used to assess changes in hemodynamic activity in the PFC during single-task and dual-task gait, with and without obstacles; obstacles were actually holograms which were presented as red shapes using lasers (four tasks in all). Slow gait moderated the effect of the obstacles on changes in hemodynamic activity across task conditions. The levels of hemodynamic activity were higher in a dual-task than a single-task gait, regardless of the presence of obstacles.

Wagshul et al. [54] applied structural MRI and fNIRS to analyse brain activation during dual-task gait and the effect of brain volume on the performance of healthy older people. They found that the loss of grey matter in healthy older adults can lead to excessive activation of the frontal lobe during cognitively demanding walking tasks while the behavioural changes were not observed. This provides evidence for the concept referred to as neural inefficiency.

Salzman et al. [55] investigated the neural mechanisms underlying the interactions between gait performance, age, and deficits in executive function and motor control during stair climbing. Their study involved 20 men and women (mean age 73 years) from a community in Canada. Participants completed the single tasks (standing and responding to a response-time task; ascending or descending stairs) and a dual task (responding while ascending or descending stairs). Hemodynamic response and deoxyhaemoglobin changes in the PFC were assessed using fNIRS. Gait performance, including vocal response time and accuracy, was recorded using smart shoe insoles. Executive processes were found to be less efficient during dual tasks than single tasks, suggesting that neural changes may precede gait performance declines, as no significant reduction was observed in gait speed.

Salzman et al. [56] additionally examined how elderly people can alleviate the need for gait performance when performing cognitively demanding dual tasks. Twenty healthy older adults (mean age 72 years) were requested to perform cognitively demanding tasks, in increasing order of difficulty, while walking. During the tasks, the hemodynamic response and deoxyhaemoglobin changes in the PFC were assessed using fNIRS, and gait speed, response time, and accuracy were recorded. The research revealed that participants used an automatic motor control strategy to reduce the need for executive function; shifting conscious attention from walking during a dual task resulted in a decrease in the haemodynamic response and deoxyhaemoglobin changes in the PFC. However, the authors concluded that reducing prefrontal activation is inefficient in maintaining response time and accuracy and may be reduced further by greater cognitive demands.

### 4.2. fNIRS, Dual-Task Gait, Neurological Disorders

Pelicioni et al. [57] systematically reviewed fNIRS studies to assess the results of PFC activation during dual tasks and the effects of age and clinical diseases/disorders, identifying 35 studies that met inclusion criteria. Increased PFC activation was commonly found in studies involving dual tasks. Regardless of the type of secondary tasks performed during walking, study groups with clinical diseases generally showed increased PFC activation, indicating that these individuals need more neurological resources to walk safely. A combined systematic literature review [58] (35 studies) and meta-analysis (17 studies), performed in 2021 by Bishnoi et al., investigated the feasibility of using fNIRS to measure the activation in the PFC changes during single-task and dual-task gait in healthy young adults, healthy older adults, and people with Parkinson’s disease, stroke, and multiple sclerosis. The study identified differences between dual- and single-task gait in persons with multiple sclerosis and stroke. Older adults with neurological disease showed increased brain activation during dual-task gait. These recent review papers collectively gathered valuable information about fNIRS on age-related and neurological-disease-related prefrontal activation, confirming that fNIRS is a promising tool for investigating adverse cognitive and motor outcomes in ageing populations with neurological conditions.

### 4.3. fNIRS, Dual-Task Gait, Cognitive Dysfunction, Dementia

Teo et al. [59] investigated brain activation in older adults with or without subjective memory complaints and dementia during gait tasks. This study included 58 men and women (23 with subjective memory complaints, 9 with dementia, 26 healthy) aged 65–94 years. Gait performance (stride velocity and length) was measured using an electronic walkway. Changes in oxyhaemoglobin in the left PFC during gait were measured using fNIRS. This study reported that, in a single-task gait test, a greater increase in oxyhaemoglobin was observed in those with dementia than in the other two groups. A significant increase in oxyhaemoglobin was observed during dual-task performance in the subjective memory complaint group, but a significant decline was observed in the dementia group. The authors speculated this could be due to a failure of individuals with dementia to allocate adequate attentional resources to the increasing cognitive demands of the dual task. However, major limitations were acknowledged by the authors including their small sample size and the use of a single-channel fNIRS system, highlighting that multi-channel fNIRs systems capable of measuring other key brain regions should be utilised.

## 5. Brain–Drive Hypothesis

Early AD is characterised by impaired memory and learning, reflecting damaged and destroyed synapses [60] and the death of healthy neurons. The neuropathological characteristics of AD are the co-presence of extracellular amyloid plaques and intraneuronal neurofibrillary tangles, due to the accumulated protein of amyloid-β (Aβ) peptides for plaques and toxic tau for tangles, and that the soluble forms of Aβ and tau work together or independently [60]. Muurling et al. [61] examined differences in gait between participants with mild cognitive impairment (n = 58), dementia (n = 26), and healthy controls (n = 58), the relationship between gait and cerebrospinal fluid biomarkers, and whether gait is a predictor of cognitive decline. Participants were asked to perform single- and dual-task gait tests; performance was measured and analysed using wearable devices. Gait disturbances were found to be associated with cognitive impairment and cerebrospinal fluid tau levels [61]. A longitudinal study by Wennberg et al. [62] of 439 healthy participants aged 50–69 years examined the association between cerebral Aβ and changes in gait, and the mediating effect of cortical thickness. Gait parameters were assessed using an electronic walkway. Cerebral Aβ deposition was assessed by Pittsburgh Compound B (PiB)–PET and cortical thickness using 3T MRI. Follow-up was conducted for a median of 15.6 months. This study reported that higher brain Aβ was associated with faster gait decline in women and that cortical thickness did not mediate this association.

Nadkarni et al. [63] studied 183 older adults without dementia (mean age 86, 58% men, 97% Caucasian, 144 cognitively normal), finding that cerebral Aβ deposition was associated with slower gait speed and cognition status and that apolipoprotein E (APOE) ε4 mediated the association. Whitson et al. [64] examined dual-task gait performance among 29 cognitively normal older adults aged 60–72 years, 14 with and 15 without an APOE ɛ4 allele (associated with higher risk of AD). Walking and cognitive tasks were performed independently and concurrently. The study revealed that subtle changes in brain function were reflected in dual-task gait performance in older adults who had an APOE ɛ4 allele but lacked AD symptoms.

## 6. Research Gap and Future Directions

Cognitive frailty should be approached from a new perspective, using cutting-edge techniques that detect subtle differences and methods for identifying neurophysiological markers that might link physical and cognitive deterioration. Functional NIRS has not been applied in the field of cognitive frailty; markers of decline in both cognitive function and physical performance have not been identified in conjunction with these measures in a large-scale, population-based study. Future studies are recommended to address this highly novel and important area of inquiry that will provide a significant step towards addressing the growing problem of cognitive decline and dementia, for which there is currently no effective cure or preventive strategy.

## 7. Conclusions

This review examined existing evidence for the potential to develop and use an fNIRS-based screening tool for dementia in population-based studies. fNIRS is motion-tolerant and not subject to environmental constraints; it is a promising tool for the identification of ageing markers and for quantifying physiological adaptations following anti-ageing interventions. The existing literature provides encouraging evidence that fNIRS may be useful as a diagnostic screening technology for investigating the human cortex as a target for screening for dementia risk years before onset because it can detect subtle neurophysiological markers of cognitive dysfunction. Therefore, fNIRS may enable a better understanding of brain activation patterns in people with cognitive dysfunction, especially during dual-task activities that require both cognitive and physical effort. Gait parameters such as speed, swing frequency, and length can be measured using computer-based tools, with participants walking along a sensor-equipped mat while gait parameters are recorded and analysed digitally. Gait control under dual-task conditions reflects attention control strategies and cognitive flexibility. Dual-task gait and gait in single-assessment tasks are often compared in order to distinguish neurological disorders with shared neuropathology [41]. Understanding the neural mechanisms that underpin dual deficits in gait and brain function is critical for optimising exercise and cognitive training interventions. The neurophysiology of human cognitive ageing, informing novel therapeutic targets for boosting the efficacy of lifestyle interventions that delay dementia onset, will provide the evidence base needed to co-design interventions for delaying dementia—notably, a screening tool for early detection of dementia risk. The findings can shape strategies for preventing or slowing cognitive frailty in older people through the development of tailored exercise, nutrition, and cognition interventions that improve both physical health and brain health.

## Figures and Tables

**Figure 1 brainsci-12-00968-f001:**
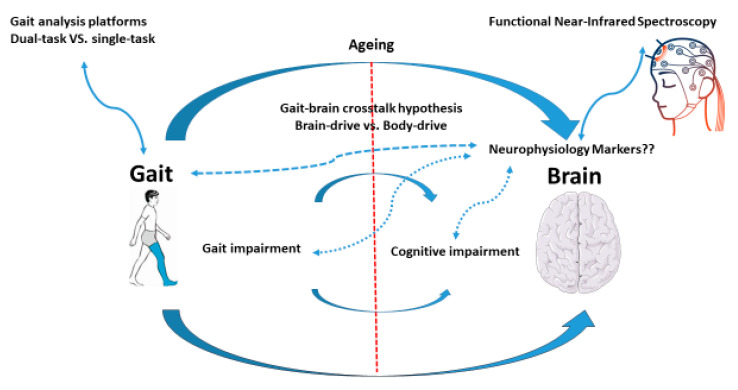
Model of dual gait and brain function decline. The Figure was partly generated using Servier Medical Art, provided by Servier, licensed under a Creative Commons Attribution 3.0 unported license; also from https://brainvision.com/applications/nirs/. Accessed on 22 April 2022.

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
