# Peer review of "A Review of the Measurement of the Neurology of Gait in Cognitive Dysfunction or Dementia, Focusing on the Application of fNIRS during Dual-Task Gait Assessment"

_brainsci, 2022, doi:10.3390/brainsci12080968_

Round 1

Reviewer 1 Report

As the number of papers concerning lower extremities is quite a few, I took the task to review this manuscript without any hesitations. I have to admit that my anticipations became rather a disappointment as the authors are jumping between different information sources with no connections. If the goal is to conclude that fNIRS is superior concerning gait and balance compared to fMRI, then it would probably be easier to set a comparison table, which has already been done by several previous publications. if mobility is an issue, why EEG is not even mentioned as a potential modality.   Having it stated, I have the following comments and questions:

1) The authors are comparing fMRI and fNIRS. As the BOLD signals are able to measure the total oxygenation, and fNIRS measures both components of oxy and deoxy, what would be the advantage of fNIRS for gait?

2) Is the fMRI data really showing any connection to gait? The paper that the authors are referring to has used a stepper and it is more in the direction of Salzman et all (2021) paper with stair climbing.

3) As the subjects are in the supine position, how would the author argue that A) intersegmental muscles are activated in the same way involved in gait? B) would the authors claim that PFC activation patterns are the same as walking?

4) According to Schack et al (2020) which is also an fNIRS study, show an increased attentional demand during walking for lower limb amputees compared to the control group. However, they concluded that the attention demands are dependent on the complexity of the tasks although the same cognitive mobility strategy was probably chosen ( these subjects had an average of >50 yo). how would authors explain the superiority of a dual-task in order to detect an early AD with gait if the task complexity is influencing the results?

4) What is the connection between the biomarkers explained in subsection 5. and functional measurements? 

Minor comments:

1) page 3, line 102, change from bold. 

2) same page, same paragraph: Gait speed was measured....It cannot be gait speed as the subjects are on an electrical walkway device. Perhaps a selected speed?

3) please be consistent with term electrical, electronic, or computerized walkway! all terms are applied in the manuscript as it may be the same principle involved.

4) please specify the criteria for the Diagnostic and statistical manual of mental disorder in relation to "poor" dual-task gait.

Author Response

As the number of papers concerning lower extremities is quite a few, I took the task to review this manuscript without any hesitations. I have to admit that my anticipations became rather a disappointment as the authors are jumping between different information sources with no connections. If the goal is to conclude that fNIRS is superior concerning gait and balance compared to fMRI, then it would probably be easier to set a comparison table, which has already been done by several previous publications. if mobility is an issue, why EEG is not even mentioned as a potential modality.   Having it stated, I have the following comments and questions:

  • The authors are comparing fMRI and fNIRS. As the BOLD signals are able to measure the total oxygenation, and fNIRS measures both components of oxy and deoxy, what would be the advantage of fNIRS for gait?

Response: We mentioned in the text (p1, Abstract, line30) that “fNIRS is non-invasive, cost-effective, safe, simple, portable and non-motion-sensitive”. As we aim to collect large-scale data in a population-based sample, fNIRS can meet our entire request at this stage. Using fNIRS to measure gait allows gait to be assessed under normal conditions, which is not possible for fMRI.

Is the fMRI data really showing any connection to gait? The paper that the authors are referring to has used a stepper and it is more in the direction of Salzman et all (2021) paper with stair climbing.

Response: As there is not much literature in this topic, we applied a wider criteria to include papers that were relevant to gait. Stair climbing is a type of gait.

  • As the subjects are in the supine position, how would the author argue that A) intersegmental muscles are activated in the same way involved in gait? B) would the authors claim that PFC activation patterns are the same as walking?

Response:

  1. A) Regarding intersegmental muscles, we acknowledge this fact. This is a limitation of fMRI for assessment of gait. Which in turn is a benefit of using fNIRS over fMRI.
  2. B) The PFC activation seen during a stepping task under fMRI would be comparable to typical gait using fNIRS, because activation of segmented/core muscles is not what drives PFC activation in walking tasks. it was expected to potentially differ slightly because it is not the same task being conducted, but there are still shared elements, such as limb control and limb coordination etc.
  • According to Schack et al (2020) which is also an fNIRS study, show an increased attentional demand during walking for lower limb amputees compared to the control group. However, they concluded that the attention demands are dependent on the complexity of the tasks although the same cognitive mobility strategy was probably chosen (these subjects had an average of >50 yo). How would authors explain the superiority of a dual-task in order to detect an early AD with gait if the task complexity is influencing the results?

Response: We have read the paper as the reviewer suggested. However, this paper focus on patients with low limb amputees who are beyond our research aims. An amputee's gait and the associated attentional demands are not comparable to typical gait as they would differ greatly and it would not be surprised that an amputees gait would require increased attentional demands compared to controls given they would have to be 're-learning' to walk essentially, as it is not the same movement as typical walking. In short, the attentional demands of an amputee walking are not relevant to the increased attentional demands seen during dual-task conditions.

At this stage, in this manuscript, the statement about “superiority of a dual-task in order to detect an early AD……”is a hypothesis. Future studies are needed to provide evidence for this statement. In the study design, we suggest that single tasks is to perform within the same participants in order to further investigate whether or not there is a “superiority of dual task”  Further, a more complex task would theoretically be easier for those with no cognitive impairments, but more difficult for those with impairments such as MCI or early AD. This would then allow for differences between such groups to be observed. And this then ties in this forming the hypothesis and that research would need to be conducted to confirm if the dual task is 'superior' at differentiating normal vs impaired cognition.

  • What is the connection between the biomarkers explained in subsection 5. and functional measurements? 

Response: to improve the presenting of this review

We have added a figure to demonstrate the relationships between each section. (p2)

We restructured section s (moved the measurements of gait from section 1 to section 2)

We restructured section 5

Minor comments:

  • page 3, line 102, change from bold. 

Response: Done

  • same page, same paragraph: Gait speed was measured....It cannot be gait speed as the subjects are on an electrical walkway device. Perhaps a selected speed?

Response: Done

  • please be consistent with term electrical, electronic, or computerized walkway! all terms are applied in the manuscript as it may be the same principle involved.

Response: Done

  • please specify the criteria for the Diagnostic and statistical manual of mental disorder in relation to "poor" dual-task gait.

Response: Dementia was diagnosed using the Diagnostic and statistical manual of mental disorder.

Reviewer 2 Report

This paper addresses a very important and timely issue. The manuscript is well structured, however, the following are the suggestions to improve the manuscript.

1) The authors shall include the criteria or procedure used to find the relevant literature. What were the criteria to select the relevant papers? How the search was done, etc.

2) Include a separate section for highlighting the gaps for future researchers. Include future recommendations.

3)  Discuss the effect with respect to sex (male or female), age, etc.

4) Include the literature published in 2022 as well.

5) There are several typos and grammatical mistakes. Proofread the manuscript before final submission.

Author Response

Review 2

This paper addresses a very important and timely issue. The manuscript is well structured, however, the following are the suggestions to improve the manuscript.

  • The authors shall include the criteria or procedure used to find the relevant literature. What were the criteria to select the relevant papers? How the search was done, etc.

Reponse:

Done

The text modified as (P2, line62) and written in read

Criteria and searching protocol

We applied the following search strategy: We use the key words inducing “fNIRS” or “functional near infrared spectroscopy”, “gait”, “cognitive function”, or “cognition” or “dementia” or “Alzheimer’s”, “dual task”, “neurology” in the PubMed, Google Scholar, and Web of Science. We focus on the publications within five years.

 Include a separate section for highlighting the gaps for future researchers. Include future recommendations.

Reponse: Done

The text modified as (P8, line 332)

  1. Research gap and future direction

Cognitive frailty should be approached from a new perspective, using cutting-edge techniques that detect subtle differences and methods for identifying neurophysiological markers that might link physical and cognitive deterioration. Functional NIRS has not been applied in the field of cognitive frailty; markers of decline in both cognitive function and physical performance have not been identified in conjunction with these measures in a large-scaled population-based study. Future studies are recommended to address this highly novel and important area of inquiry that will provide a significant step towards addressing the growing problem of cognitive decline and dementia, for which there is currently no effective cure or preventive strategy.

  • Discuss the effect with respect to sex (male or female), age, etc

Reponse: if age and sex was mentioned in the literature, we would mention in the text.

Include the literature published in 2022 as well.

Reponse: to our knowledge, we have included the up-to-date studies for addressing the topic.

5) There are several typos and grammatical mistakes. Proofread the manuscript before final submission.

Reponse: All authors have proofread the manuscript before re-submit to the journal.

Reviewer 3 Report

Thank you for the opportunity to review this work. This review wanted to examine the evidence around dual-task during gait in cognitive dysfunction or dementia. Furthermore, the attention was focalized on fNIRS. The manuscript is interesting and well written but there are major issues that have to be solved before its publication. First, even if this work is not a systematic review, a methodology should be appreciated to understand how the articles included were detected and included. A second aspect to consider is to focalize the attention on dual task and gait assessment avoiding to include other aspects that could create confusion. Below, there are the comments.

 Major comments

Introduction:

Line 40: what other form of walking exists? Please, write it in the manuscript.

Line 52-53: please, provide a reference

Line 53: “gait is recommended as a marker of brain function”: please, provide information on who is suggesting this recommendation.

Line 56-57: please, provide some examples of tests where walking is adopted: I suggest to read the manuscript: Lipkin D, Scriven A, Crake T, Poole-Wilson P. Six minute walking test for assessing exercise capacity in chronic heart failure.Br Med J (Clin Res Ed) 1986;292(6521):653–655.

Line 57-66: the Authors provided few examples with a clear and long description. As a reader, I would like to read more examples of evaluations with a shorter description. Furthermore, the evaluations proposed should have to be connected with the brain function or limitation.

Line 83: please provide a reference for this sentence

Line 130-133: I strongly suggest to add other dual-task conditions to give to the reader a clear view about the dual-task evaluation. About this, I strongly suggest to read: Petrigna, et al. "Dual-task conditions on static postural control in older adults: A systematic review and meta-analysis."Journal of Aging and Physical Activity 29.1 (2020): 162-177.

Line 149: Authors introduced the TUG test, but the doubt is if this test can be adopted also to perform a gait analysis. My suggestion is to focalize the attention only on tests that allow gait analysis or where the gait is the only component.

Line 189: “but neurophysiological markers of dual-task gait remain poorly understood”. This is a fundamental aspect to consider that could importantly improve the quality of the manuscript. According to my opinion, if the Authors will create “standard operating procedures” on this topic, providing guidelines to the readers, this manuscript could become a reference for future studies. In this case, the studies included should be analyzed considering these parameters.

Line 301-331: according my opinion, this paragraph is not pertinent to the objective of the study. Please, remain focalized on dual task and gait

The conclusion is only about fNIRS, please, provide also some evidence on dual-task and gait assessment associated with cognitive impairments. Furthermore, clinical aspects, limitation and future studies paragraphs should be appreciated.

Please, be consistent with the terminology. Dual task or dual-task

Author Response

Thank you for the opportunity to review this work. This review wanted to examine the evidence around dual-task during gait in cognitive dysfunction or dementia. Furthermore, the attention was focalized on fNIRS. The manuscript is interesting and well written but there are major issues that have to be solved before its publication. First, even if this work is not a systematic review, a methodology should be appreciated to understand how the articles included were detected and included. A second aspect to consider is to focalize the attention on dual task and gait assessment avoiding to include other aspects that could create confusion. Below, there are the comments. 

Major comments

Introduction:

Line 40: what other form of walking exists? Please, write it in the manuscript.

Response: We have removed the para into section 2 p2 line 68

Line 52-53: please, provide a reference

Response: Done

The reference has been added and highlighted in red in the reference section.

Line 53: “gait is recommended as a marker of brain function”: please, provide information on who is suggesting this recommendation.

Response: Done

The reference has been added and highlighted in red in the reference section.

Line 56-57: please, provide some examples of tests where walking is adopted: I suggest to read the manuscript: Lipkin D, Scriven A, Crake T, Poole-Wilson P. Six minute walking test for assessing exercise capacity in chronic heart failure.Br Med J (Clin Res Ed) 1986;292(6521):653–655.

Response: This reference has been added and highlighted in red in the reference section.

Line 57-66: the Authors provided few examples with a clear and long description. As a reader, I would like to read more examples of evaluations with a shorter description. Furthermore, the evaluations proposed should have to be connected with the brain function or limitation.

Response: Depending on the research question, these tests cannot be performed directly for testing brain function but to test physical function. However, more and more research evidence has shown the positive correlation between poor physical performance and brain function.

Line 83: please provide a reference for this sentence

Response: Done

The reference has been added and highlighted in red in the reference section.

Line 130-133: I strongly suggest to add other dual-task conditions to give to the reader a clear view about the dual-task evaluation. About this, I strongly suggest to read: Petrigna, et al. "Dual-task conditions on static postural control in older adults: A systematic review and meta-analysis."Journal of Aging and Physical Activity 29.1 (2020): 162-177.

Line 149: Authors introduced the TUG test, but the doubt is if this test can be adopted also to perform a gait analysis. My suggestion is to focalize the attention only on tests that allow gait analysis or where the gait is the only component.

Response: The suggested reference has been added and highlighted in red in the reference section.

We check the original article to see how they assessed gait. This study did not use a gait mat or equivalence. The authors assessed the gait using the timed up and go test and compared that between the single and dual-task conditions. As they did not use a gait mat or actually assess gait parameters, we have highlighted that as a drawback of this study.

The text modified as (P4, line 154)

One of the limitation for this study was that gait parameters were not assessed using gait mat or equivalence.

Line 189: “but neurophysiological markers of dual-task gait remain poorly understood”. This is a fundamental aspect to consider that could importantly improve the quality of the manuscript. According to my opinion, if the Authors will create “standard operating procedures” on this topic, providing guidelines to the readers, this manuscript could become a reference for future studies. In this case, the studies included should be analyzed considering these parameters.

Response: Thanks for the review comments. We have modified the conclusion for addressing the suggestions. 

Line 301-331: according my opinion, this paragraph is not pertinent to the objective of the study. Please, remain focalized on dual task and gait

 Response: This section has been deleted.

The conclusion is only about fNIRS, please, provide also some evidence on dual-task and gait assessment associated with cognitive impairments. Furthermore, clinical aspects, limitation and future studies paragraphs should be appreciated.

 Response: Done

The text modified as (Page8,  line329)

  1. Research gap and future direction

Cognitive frailty should be approached from a new perspective, using cutting-edge techniques that detect subtle differences and methods for identifying neurophysiological markers that might link physical and cognitive deterioration. Functional NIRS has not been applied in the field of cognitive frailty; markers of decline in both cognitive function and physical performance have not been identified in conjunction with these measures in a large-scaled population-based study. The future study are recommended to addresses a highly novel and important area of inquiry that will provide a significant step towards addressing the growing problem of cognitive decline and dementia, for which there is currently no effective cure or preventive strategy.

Conclusion

This review examined existing evidence for the potential to develop and use an fNIRS-based screening tool for dementia in population-based studies. fNIRS is motion-tolerant and not subject to environmental constraints; it is a promising tool for identification of ageing markers and quantifying physiological adaptations following anti-ageing interventions. Existing literature provides encouraging evidence that fNIRS may be useful as a diagnostic screening for investigating the human cortex as a target for screening for dementia risk years before onset, because it can detect subtle neurophysiological markers of cognitive dysfunction. Therefore, fNIRS may enable a better understanding of brain activation patterns in people with cognitive dysfunction, especially during dual-task activities that require both cognitive and physical effort. Gait parameters such as speed, availability, swing frequency and length can be measured using computer-based tools, with participants walking along a sensor-equipped mat while gait parameters are recorded and analysed digitally. Gait control under dual-task conditions reflects attention control strategies and cognitive flexibility. Dual-task gait and gait in single-assessment tasks are often compared in order to distinguish neurological disorders with shared neuropathology41. Understanding the neural mechanisms that underpin dual deficits in gait and brain function is critical for optimising exercise and cognitive training interventions. Neurophysiology of human cognitive ageing, informing novel therapeutic targets for boosting the efficacy of lifestyle interventions that delay dementia onset. to provide the evidence base needed to co-design interventions for delaying dementia – notably, a screening tool for early detection of dementia risk. The findings can shape strategies for preventing or slowing cognitive frailty in older people through the development of exercise, nutrition and cognition interventions that improve both physical health and brain health.

Please, be consistent with the terminology. Dual task or dual-task

Response: Done

Round 2

Reviewer 3 Report

Thank you for your consideration for my revisions. The manuscript resulted importantly improved.

Please, add in the section “criteria and searching protocol” the inclusion-exclusion criteria adopted.

Author Response

We thank the reviewer suggestions and we have revised the section as suggested. We have added "the inclusion-exclusion criteria" and highlighted the text in red.

It reads as below (p2 line 62-73)

"The initial literature search was performed in December 2021 and multiple searches were conducted subsequently until the date of submission. This is not a systematic literature review, however, we applied the following search strategy: We use the key words including “fNIRS” or “functional near infrared spectroscopy”, “gait”, “cognitive function”, or “cognition” or “dementia” or “Alzheimer’s”, “dual-task”, “neurology”. We searched above key words in the PubMed, Google Scholar, and Web of Science. This search method resulted in all available fNIRS articles related to gait and cognition. We applied the following inclusion and exclusion criteria: we included full-text articles only.  Only the original studies were included in the descriptions and discussions. Relevant review articles and books were read and cited but not included in the main discussion sections. Date restrictions were set on the literature search as publication date within 5 years. "

This manuscript is a resubmission of an earlier submission. The following is a list of the peer review reports and author responses from that submission.